# A Review of the Strategic Use of Sodium Alginate Polymer in the Immobilization of Microorganisms for Water Recycling

**DOI:** 10.3390/polym16060788

**Published:** 2024-03-12

**Authors:** Yaneth A. Bustos-Terrones

**Affiliations:** CONAHCYT-TecNM-Technological Institute of Culiacan, Division of Graduate Studies and Research, Culiacan 80220, Mexico; yaneth.bt@culiacan.tecnm.mx

**Keywords:** immobilization of microorganisms, sodium alginate, wastewater treatment, contaminants

## Abstract

In the quest for advanced and environmentally friendly solutions to address challenges in the field of wastewater treatment, the use of polymers such as sodium alginate (Na-Alg) in combination with immobilized microorganisms (IMs) stands out as a promising strategy. This study assesses the potential of Na-Alg in immobilizing microorganisms for wastewater treatment, emphasizing its effectiveness and relevance in environmental preservation through the use of IMs. Advances in IMs are examined, and the interactions between these microorganisms and Na-Alg as the immobilization support are highlighted. Additionally, models for studying the kinetic degradation of contaminants and the importance of oxygen supply to IMs are detailed. The combination of Na-Alg with IMs shows promise in the context of improving water quality, preserving ecological balance, and addressing climate change, but further research is required to overcome the identified challenges. Additional areas to explore are discussed, which are expected to contribute to the innovation of relevant systems.

## 1. Introduction

The rapid increase in population and the intensification of human activities have raised growing concerns regarding water pollution [1,2]. Sustainable water resource management has become essential as the excessive accumulation of contaminants poses a significant threat to aquatic and terrestrial ecosystems, resulting in considerable adverse impacts on water quality, biological diversity, and public health [3,4,5,6].

Despite advances in conventional water treatment methods, there still exist various limitations that require careful attention [7,8,9]; notable challenges include the generation of large volumes of wastewater, its costly management, high energy consumption, and environmental risks associated with the use of chemicals [10,11]. The susceptibility of water treatment methods to variations in water conditions has driven the search for more advanced and sustainable approaches [12,13]. In this context, the quest for innovative and strategic treatments has become a necessity. One approach currently being employed is using IMs in solid matrices [14,15].

These IMs include bacteria, fungi, and other microbes, which are bound or confined within solid support structures [16,17]. This unique configuration allows the microorganisms to remain in specific locations, carrying out their contaminant removal functions without the need to move freely through the aquatic environment [4,18]. Compared to free-living microorganisms, IMs play a significant role in wastewater treatment, offering greater benefits in terms of efficiency and biotechnological applications. They enable processes with longer retention times, enhance efficiency in contaminant degradation, reduce the generation of residual sludge, and simplify product recovery, and they have been proven to be highly cost-effective [10,19].

The immobilization of microorganisms using biomaterials has emerged as a revolutionary strategy, aiming to harness the potential of microbial communities for efficient and environmentally friendly wastewater treatment [20]. Among many biomaterials, sodium alginate—derived from brown algae—has emerged as a prominent candidate in this context [5]. Furthermore, the biocompatibility and biodegradability of Na-Alg further highlight its appeal as a biomaterial for immobilization applications [21]. These polymers (Na-Alg) provide a solid matrix for trapping microorganisms, protecting them from adverse environmental conditions, regulating the diffusion of nutrients and products, facilitating their recovery and re-use, reducing waste generation, and adapting to different applications and types of microorganisms [1,2].

This review aims to explore the interactions between Na-Alg and microorganisms, delving into the nuances of the role of Na-Alg as an immobilization matrix in wastewater treatment processes. The exploration spans a wide range of studies, from laboratory-scale experiments to real-world applications, with the goal of unraveling the full potential of Na-Alg. Through examining recent findings, methodological approaches, and the results of sodium alginate-based immobilization, this review seeks to provide a comprehensive understanding of the current state of knowledge regarding the utilization of this biomaterial for wastewater treatment. The integration of microorganism immobilization techniques using sodium alginate is a promising avenue in the pursuit of sustainable and effective wastewater treatments, showcasing a strategic and environmentally friendly approach to address the challenges posed by water pollution.

## 2. Materials for Microorganism Immobilization

The choice of material used for immobilizing microorganisms is crucial as it influences their viability and the effectiveness of wastewater treatment [22,23]. Three types of supports are available: organics—which can be further divided into natural (e.g., chitin, agar, alginate, and carrageenan) and synthetic organics (e.g., polyvinyl alcohol, PVA; polyvinylpyrrolidone, PVP; polypropylene amine; polyurethane; and acrylamide)—and inorganics (e.g., activated carbon, clay, zeolite, anthracite, ceramics, and porous glass) [22,24]. Of these materials, alginate is the most widely used support in water treatment as it forms microspheres when mixed with microorganisms and calcium [19].

Some researchers [22] have pointed out that polymers offer high diffusion rates and are environmentally friendly, stable in wastewater samples, durable and mechanically resistant, and, most importantly, non-toxic to microorganisms. Therefore, various studies [25,26] have employed polymers as supports to immobilize microorganisms, with Na-Alg mainly being used. However, other researchers have chosen to use polyurethane foam [27], activated carbon [28], zeolites [29], silica gels [30], and so on. Mehrotra et al. [31] mentioned that microorganisms immobilized in hydrogels exhibit greater tolerance to toxic contaminants due to protective encapsulation. Nevertheless, several researchers have opted to use Na-Alg due to its specific characteristics. The deliberate selection of support materials remains a critical aspect in optimizing wastewater treatment processes with the goal of balancing factors, such as toxicity, durability, and mechanical resistance, to ensure the success of microorganism immobilization techniques.

## 3. Intrinsic and Specific Properties of Polymers

The intrinsic properties of polymers, which are linked to their molecular structures, are crucial for their use in specific applications. These properties also play key roles in improving water quality and conservation [32]. Due to their adsorption and filtration capabilities, polymers can retain contaminants, while their abilities to control their hydrophobicity/hydrophilicity facilitate their interaction with water. Moreover, features such as ionic exchange, chemical resistance, biocompatibility, and property modification make them effective tools for ensuring sustainable performance in water treatment [33]. The formation of hydrogels, contaminant detection, durability, mechanical strength, and compatibility with other treatment methods are some of the factors promoting the versatility of polymers in addressing water-related challenges. The versatility displayed by polymers in addressing various challenges highlights the need for continued exploration and utilization of their properties in order to enhance water treatment processes and promote environmental sustainability.

The extraction of alginate from seaweed involves breaking down the cell walls and releasing alginate. This process may involve the hydrolysis of glucosidic bonds and the esterification of carboxyl groups. Subsequently, in the precipitation process, calcium ions, which are commonly derived from calcium chloride, react with the sodium ions present in the alginate, forming complexes that precipitate Na-Alg. Sodium alginate is a linear copolymer of β-D-mannuronic acid (M) residues, linked with α-L-guluronic acid (G) through 1,4-glycosidic bonds. The residues can form a homopolymer composed solely of one type of monosaccharide or heteropolymer blocks constructed from both types [34,35,36,37]. Alginates are natural polysaccharides produced by bacteria or extracted from brown algae, which consist of β-D-mannuronate (M blocks) and α-L-guluronate (G blocks) linked with 1→4 bonds. These bonds alternate between mannuronate–guluronate structures and guluronate–mannuronate dyads (see Figure 1) [38,39]. The properties of Na-Alg influence viscosity, which varies depending on various biological, ecological, and process factors. Figure 2 presents the procedure for the synthesis of Na-Alg [7].

### Uses and Properties of Sodium Alginate

Sodium alginate, a polysaccharide extracted from seaweed, is notable for its ability to form gels with calcium ions. This biopolymer derived from brown algae is non-toxic, biodegradable, and water-soluble, aligning with the principles of green chemistry. Its versatility is evident through its use in the food, pharmaceutical, and cosmetic industries. Additionally, it is considered safe for medical applications and has been employed in biomedical products and controlled drug release matrices [40,41]. Its stability across various pH levels and its water retention capacity make it valuable for use in coatings and thickeners across diverse industries [42,43]. Characterized by low toxicity, biocompatibility, and biodegradability, alginate is a natural polymer with gelation and thickening properties and is renowned for its cost-effectiveness [44,45,46]. Its ionotropic gelation ability upon contact with cations is particularly noteworthy [47].

Sodium alginate polymers have unique advantages for water treatment due to their ability to form gels and their biocompatibility. These properties make Na-Alg effective in immobilizing microorganisms and retaining contaminant substances in aqueous solutions (see Figure 3). Due to their advantageous properties, alginates have been extensively utilized in the pharmaceutical industry as they can form gels through cross-linking with calcium ions (ionotropic gelation process). Furthermore, as they are non-toxic and non-immunogenic compounds, they have been employed in the pharmaceutical industry for drug or enzyme encapsulation [34,48]. On the other hand, Na-Alg is also used in the textile industry due to its ability to form coatings, enhance the properties of textiles, and contribute to processes such as printing and fabric finishing [49,50]. Similarly, due to its binding, thickening, gelation, film-forming, and stabilizing properties, Na-Alg has been widely used in the food industry [40,51]. Sodium alginate is primarily employed as an efficient material in biotechnology, specifically for the immobilization of cells and enzymes [13,21,52]. Novel forms of microorganism immobilization using Na-Alg are currently being explored in order to obtain various advantages such as enhanced process control, improved biocatalytic activity, and extended process time intervals [53,54]. Microorganisms immobilized in alginate gel are being used to promote new strategies, such as preventing enzymes from leaching into the medium and facilitating the external release of products [55,56,57].

## 4. Sodium Alginate as an Adsorbent

The versatility of Na-Alg has been demonstrated through its ability to be chemically modified, allowing for adaptation to specific treatment conditions. Sodium alginate stands out as an effective adsorbent for use in water treatment. Its unique capacity to form gels in the presence of calcium ions allows for the creation of hydrogel matrices, facilitating its use as an adsorbent [58,59,60,61,62] or filtration medium [46]. Its effective performance, applicability for contaminant removal, and use in hydrogel matrices for contaminant adsorption make it a valuable component in environmental management and water treatment. Various researchers have highlighted the versatility and efficacy of this polymer in environmental applications [63,64,65,66].

Gao et al. [67] provided a comprehensive review focused on the development of sodium alginate-based adsorbents for heavy metal removal. On the other hand, Banerjee et al. [68] revealed the influence of alginate concentration when used as an adsorbent for nutrient removal. Phiri et al. [69] removed cationic, anionic, and organic pollutants from highly acidic water using alginate beads.

Kamaci and Kamaci [59] explored the capacity of Na-Alg in terms of the adsorption of textile dyes, such as methylene blue. Other researchers have used Na-Alg in combination with catalysts to enhance the adsorption process [38]. For example, Lu et al. [70] utilized sodium alginate/polyvinylpyrrolidone hydrogels; Hosseini et al. [71] performed adsorption with Na-Alg in combination with biosilicate/magnetite nanocomposite for the removal of pesticide malathion; and Khiavi et al. [72] combined Na-Alg with silica nanoparticles for aflatoxin reduction. Together, these studies highlight the promising applicability of Na-Alg in the context of environmental treatment, demonstrating its ability to address various challenges associated with the removal of aqueous contaminants. These diverse and extensive research findings underscore the adaptability of Na-Alg as a versatile and effective adsorbent in addressing a wide spectrum of environmental challenges. Table 1 summarizes existing studies focused on the use of Na-Alg as an adsorbent, while Table 2 details the conditions used for immobilizing microorganisms with Na-Alg.

Studies addressing the use of Na-Alg in wastewater treatment have been published; however, most of them have focused on adsorption [5]. The author believes that it is important to expand research on the incorporation of polymers as an effective means to host microorganisms, thereby enhancing the contaminant degradation capacity. Additionally, it would be valuable to explore the simultaneous use of various non-toxic catalysts for microorganisms, aiming to increase the efficiency of pollutant degradation. As the polymer can absorb pollutants, this property could be leveraged to provide a favorable environment for the microorganisms. These microorganisms, in turn, can consume organic matter, nutrients, and other contaminants present in the water. The introduction of non-toxic catalysts could complement this process through oxidizing contaminants that are not degraded by the microorganisms, contributing to a more comprehensive and effective strategy for wastewater treatment and water resource sustainability. However, the use of Na-Alg in immobilizing microorganisms for wastewater treatment still faces crucial challenges related to optimizing the treatment efficiency and adaptability of IMs.

## 5. Immobilization Methods

The use of an incorrect immobilization method can compromise the viability, metabolic activity, and efficacy of microorganisms. As such, it is crucial to carefully select the immobilization method to ensure that the microorganisms can maintain their functionality effectively [98,99,100]. The selection of a method depends on the specific needs of the process, the characteristics of the microorganisms, and the environmental conditions [19]. Immobilization methods are classified as either physical or chemical, both offering specific advantages [101].

Physical immobilization methods involve retaining microorganisms in solid matrices or supports without chemically modifying their structures [102,103]. These methods rely on the ability to trap and maintain microorganisms in place, allowing them to grow and perform their metabolic functions, including encapsulation in gel, immobilization in porous membranes, fixation on synthetic fibers, and adherence to magnetic particles, in a confined environment [104]. On the other hand, chemical methods, such as cross-linking and covalent bonding, involve the formation of chemical bonds between microorganisms and the support (Figure 4). Both methods allow for high efficiency in the degradation of organic contaminants and nutrients in wastewater, improving the quality of the treated water [105,106]. Therefore, the selection of an appropriate method is essential to achieve successful results in water treatment processes.

## 6. Types of Immobilized Microorganisms

IMs have been employed in a wide variety of applications due to their stability, re-usability, efficiency, and process control [19,107]. An example of the use of IMs is the participation of nitrifying and denitrifying bacteria for the conversion of ammonia to nitrate and the subsequent reduction of nitrate to gaseous nitrogen, respectively. Additionally, both aerobic and anaerobic bacteria may be utilized, encompassing various strains that are capable of decomposing organic compounds under both aerobic and anaerobic conditions [108]. Other commonly used microorganisms include fungi and yeasts [109,110], microalgae [111,112], and other microorganisms (e.g., autotrophs) [113].

IMs are selected based on the specific characteristics of the wastewater and treatment objectives [114]. However, some researchers prefer to use activated sludge for immobilization due to the diversity of aerobic microorganisms and its conducive environment for contaminant degradation in wastewater. Additionally, its adaptability to various environmental conditions makes it versatile and suitable for use in various wastewater treatment systems [1,4]. Several studies have addressed the application of microorganism immobilization technology in different fields. For example, De Souza et al. [115] immobilized cells in alginate, demonstrating the high efficiency of the process; López-Menchero et al. [116] studied the immobilization of yeast in alginate biocapsules; and Xue et al. [117] investigated diesel degradation using cells immobilized in alginate, achieving a high degradation rate. These studies showcase the versatility and effectiveness of cell immobilization techniques in biotechnological applications. Likewise, the adaptability of IMs to diverse environmental conditions has contributed to the development of innovative solutions for wastewater treatment and environmental management.

### Factors Affecting Immobilized Microorganisms

The factors influencing microorganism immobilization are diverse and crucial for the success of biotechnological processes. These include the nature of the microorganism, the type of support, environmental conditions, concentration and type of microorganisms, presence of inhibitors, density loads, immobilization technique, and microorganism selection [111]. Similarly, factors influencing the activity of microorganisms include porosity, surface chemistry, ion exchange capacity, and the structural stability of the support. These elements must be optimized to ensure better efficiency and process stability [104]. Care should be taken regarding certain factors (e.g., toxic substrates, pH and/or temperature changes, inadequate oxygen concentrations, and competition among microorganisms) that may inhibit the growth and activity of IMs [22].

## 7. Contaminants Removed by Immobilized Microorganisms

Immobilized microorganisms can play a crucial role in the removal of various contaminants in wastewater treatment. Numerous studies have been conducted over the past several decades, yielding developments to improve immobilization techniques and the application of microorganisms in wastewater treatment systems. However, microorganism immobilization technology remains an active area of research and development in the field of wastewater treatment, with continuous efforts being made to enhance the efficiency and sustainability of these processes [10,13,113,118].

Microorganism immobilization has been used to address various pollution issues; for example, its use has been reported for the removal of nutrients [56,68], organic compounds [119], microplastics [4], p-nitrophenol [10,118], heavy metals [120,121], antibiotics [122], other pharmaceutical products [39,48], toxins [72], textile dyes [123,124], and petroleum [17,125], among others [25]. These studies have presented interesting results, suggesting the high efficiency of IMs in water treatment. Bustos-Terrones et al. [2] stated that the degradation of organic dyes using IMs with Na-Alg was efficient when compared to other methodologies [126,127]; however, the main difference was observed in the retention time. Table 3 lists some of the contaminants that have been removed and the associated removal efficiency achieved with IMs using Na-Alg.

### Degradation of Nutrients by Immobilized Microorganisms

Whether free or immobilized, microorganisms require essential nutrients, such as carbon, nitrogen, phosphorus, and other elements, to grow and reproduce, in addition to water, an optimal temperature, a suitable pH, and oxygen [22,133]. They may also need specific conditions regarding light, pressure, salinity, or other environmental factors depending on the species and the environment. During bioremediation, they primarily obtain nutrients from wastewater, which is typically enriched with nutrients and organic matter [31,134].

Immobilized microorganisms are effective in degrading a wide range of contaminants, primarily organic matter and nutrients. Some studies have demonstrated high removal efficiencies for nutrients and organic matter [129]. However, other studies have shown that IMs exhibit higher nutrient consumption than organic matter [113]. Shin et al. [119] reported high removals of organic matter (85%) and nutrients (99%) using IMs obtained from activated sludge. Bustos-Terrones et al. [19] reported the removal of organic matter (78.25%) and nutrients (94.26%) using microorganisms immobilized in Na-Alg. Katam and Bhattacharyya [20] revealed high efficiencies (93%) in the degradation of carbon, nitrogen, and phosphorus through the immobilization of activated sludge. Mannacharaju et al. [135] confirmed the effectiveness of degrading organic compounds (87%) using a packed-bed reactor with immobilized bacterial cells. Guo et al. [56] reported the removal of nutrients and organic matter using *Chlorella pyrenoidosa* supported in Na-Alg, with a higher nutrient removal rate (up to 81.0%). Finally, Lee et al. [129] reported a 95% removal of nutrients using immobilized *Chlorella vulgaris* and *Chlamydomonas reinhardtii* in Na-Alg.

Figure 5 illustrates the adsorption and assimilation mechanism of nutrients by IMs in Na-Alg (adapted from [68]). With respect to this figure, Banerjee et al. [68] mentioned that the adsorption of NH_4_^+^ ions relies on the electrostatic attraction between protonated amino groups (NH_3_^+^) and unprotonated carboxyl groups (COO–). Similarly, the physical interaction of PO_4_^3−^ ions and OH occurs through a hydrogen bond. The assimilation of nutrients (NH_3_^+^ and PO_4_^3−^) by microorganisms over time results in vacant positions in the alginate. This process continues until the limited nutrients are adsorbed by the microorganisms, saturating the beads with adsorbed nutrients.

## 8. Residence Times Using Immobilized Microorganisms

Residence time is an important factor that must be carefully controlled and optimized, according to the specific conditions of each process and the characteristics of the IMs, in order to maximize their effectiveness and minimize any negative effects [109,110]. IMs often exhibit long residence times, as has been reported by some researchers. A prolonged residence time can be beneficial, as it allows for greater interaction between the IMs and the substrate to be degraded; however, it can also be a disadvantage when compared to other water treatment methods.

Mannacharaju et al. [136] reported a residence time of 24 h for the removal of organic compounds using the immobilization of a microbial consortium in a fluidized reactor. In another study, Mujtaba et al. [137] reported a residence time of 48 h for the removal of nutrients and organic matter when using immobilized activated sludge. Milojkovic et al. [89] reported that 80% removal occurred at 18 h, and complete removal was achieved at 24 h. Zhang et al. [85] even stated that the maximum efficiency for IMs to degrade phenol is at 24 h. Finally, Bustos-Terrones et al. [1] mentioned that a minimum of 12 h is required for the satisfactory degradation of nutrients and organic matter. An adequate residence time can allow for desirable efficiency to be achieved in water treatment processes.

## 9. Cycles in the Use of Alginate and Immobilized Microorganisms

One of the advantages of using IMs for treatment in biopolymers is that they can be re-used in multiple cycles, as has been reported by Lu et al. [70]. The successful re-use of IMs depends on maintaining the integrity and activity of the microorganisms, ensuring the stability of the support used, and maintaining proper control of the environmental conditions during the process [1]. Wasito et al. [3] observed that *E. coli* immobilized in beads could be used for up to four cycles while still retaining a responsiveness of up to 85%, gradually decreasing in efficiency afterward. On the other hand, Sam et al. [10] mentioned that PVA/Na-Alg beads can be re-used without a significant loss in biodegradation efficiency for 20 consecutive biodegradation cycles without any leakage of biomass or disintegration. This is consistent with the findings reported by Perez et al. [26], who mentioned that the microorganisms maintained a viability of 75% after 20 cycles. Meanwhile, Zhang et al. [85] stated that 10 cycles can be performed for phenol degradation. Mujtaba et al. [137] mentioned that the efficient removal of nutrients and organic matter takes a residence time longer than 2 days, and similar results were obtained when repeating batch cycles three times. Finally, Noreen and Ahmad [138] reported that Na-Alg has the capacity to undergo a minimum of three or four adsorption–desorption cycles and a maximum of eight adsorption–desorption cycles. Therefore, IMs can be reused in biopolymer supports as long as careful considerations are made regarding their maintenance, support stability, and environmental control during successive cycles.

## 10. Degradation Kinetics of Contaminants Using Na-Alg

Kinetic studies provide crucial data for the adjustment of operating conditions and the optimization of the activity of IMs. These studies contribute to the design and improvement of biotechnological systems with various applications. The degradation kinetics of biological treatment with IMs are affected by several factors, such as the type of microorganisms used, the type of substrate, operating conditions, and the reactor design [19,68].

### 10.1. Isotherm of Adsorption

Isotherms of adsorption are used to study and describe how solutes are adsorbed on a specific surface—such as that of an adsorbent—based on the solute concentration in the liquid phase [41,59,71,86,89,117,139]. The adsorbed contaminants are determined using Equation (1).
(1)$$q_{e}=\frac{\left(C_{0}-C_{e}\right)V}{m},$$
where *q_e_* is the amount of adsorbed contaminant, *C_0_* is the initial concentration, *C_e_* is the equilibrium concentration of the solution (mg.L^−1^), *m* is the amount of adsorbent (g), and *V* is the volume of the solution (L).

The selection of the model (isotherm) depends on the specific nature of the system and the quality of the experimental data. For instance, the Langmuir isotherm model provides a mathematical representation of adsorption on a homogeneous surface with a complete monomolecular layer.

The Freundlich isotherm is a useful tool for describing adsorption in more complex and heterogeneous systems than those described by the Langmuir isotherm. These isotherms are the most used in adsorption treatments with IMs as they provide crucial information about the equilibrium between the liquid phase and the adsorbent surface during an adsorption process [89,140]. In addition to the Langmuir and Freundlich isotherms, other models, such as the Redlich–Peterson, Dubinin–Radushkevich, Sips, Toth, and Temkin isotherms, have been used (see Table 4).

### 10.2. Kinetics of Adsorption

Adsorption kinetics can be influenced by the adsorption reaction and mass transfer [148,149,150,151]. The pseudo-first-order equation (or Lagergren equation) is widely used to describe adsorption kinetics. This model assumes that the limiting step in an adsorption process is the mass transfer from the solution to the adsorbent surface and that the adsorption rate is directly proportional to the number of adsorption sites available on the adsorbent.

To study the adsorption mechanism of the contaminant on Na-Alg using the pseudo-first-order kinetic, Equation (2) may be employed [148,149,150]. On the other hand, the pseudo-second-order model is another kinetic model that is commonly used to describe adsorption kinetics. Unlike the pseudo-first-order model, this model suggests that the adsorption rate is proportional to the square of the solute concentration in the liquid phase. Equation (3) shows the pseudo-second-order kinetic model [67,71,150,151].
(2)$$Log\;\left(q_{e}-q_{t}\right)=Logq_{e}-\left(\frac{k_{1}}{2.303}\right)t,$$
(3)$$\frac{t}{q_{t}}=\left(\frac{1}{k_{2}q_{e}^{2}}\right)+\left(\frac{1}{q_{e}}\right)t,$$
where *q_e_*, *q_t_*, and *k_1_* refer to the amount adsorbed at equilibrium (mg.g^−1^); *t* is the time at which the amount of contaminate adsorbed is assessed; *k_2_* is the rate constant; and *q_e_* is the adsorption capacity in equilibrium.

## 11. Dissolved Oxygen Consumption Rate of Immobilized Microorganisms

The IMs employed for water treatment consume oxygen as part of their essential metabolic processes in order to degrade contaminants and perform necessary biological functions [152]. Through aerobic cellular respiration, IMs oxidize contaminants present in water, releasing energy for their growth and reproduction. The rate of oxygen consumption (DO) by polymer IMs can be quite specific and depends on the type of microorganism, the polymer used, and the experimental conditions. Not all IMs in the process consume the same DO concentration. For instance, Castillo et al. [153] noted that the bacterium *Azotobacter vinelandii* is characterized by a high respiration rate that is almost 10 times higher than that of *Escherichia coli*. In this context, various studies have demonstrated that adequate control of oxygen supply determines the yields and physicochemical characteristics of alginate [19,153]. For example, Bustos-Terrones et al. [19] showed that Na-Alg-IMs consumed a maximum OD of 0.75 mg O_2_/Lh, while García-Ochoa et al. [154] found that the OD concentration should be maintained above 10% of the saturation value in order to ensure that microorganisms do not die due to oxygen deficiency.

In the treatment with IMs, the oxygen transfer rate (OTR) measures the rate at which oxygen is supplied to the system, while the Oxygen Utilization Rate (OUR) indicates the rate at which microorganisms consume oxygen. These parameters are crucial for optimizing biological processes in water treatments, and their continuous monitoring allows for the adjustment of the operational conditions to ensure adequate oxygen supply and efficient system performance [154]. Ponce et al. [155] reported that they kept the OTR controlled throughout the process to maintain efficiency. Hussain et al. [156] reported an OUR value of 0.22 mg O_2_/Lh, while Bustos-Terrones et al. [1] reported an OUR value of 0.75 mg O_2_/Lh for domestic wastewater. Similar results have been reported by Bandaiphet and Prasertsan [157], who mentioned that oxygen consumption and OUR values depend on the amount of biomass. The meticulous control of oxygen supply in processes involving IMs is crucial for ensuring their efficient performance in water treatment. Likewise, the variations in oxygen consumption rates among different microorganisms highlight the importance of tailored monitoring and adjusting the operational conditions.

## 12. Case Study

A study was conducted to investigate the degradation of pollutants in household wastewater using IMs within a Na-Alg matrix, employing the entrapment method for immobilization. The microorganisms were obtained from the activated sludge tank of a wastewater treatment plant in the city of Culiacan, Mexico. These microorganisms were primarily aerobic bacteria, which are responsible for degrading nutrients in water to obtain energy and essential elements for their metabolic processes.

The sample in the laboratory was allowed to settle for 2 h, and the supernatant was subsequently separated. The initial biomass concentration was 5800 mg.L^−1^ of volatile suspended solids (VSSs). One liter of settled activated sludge was then immobilized in a 2.5% weight/volume Na-Alg solution, as illustrated in Figure 6.

The immobilization process entailed preparing a mixture using a peristaltic pump, which was subsequently dripped into a 3% weight/volume CaCl_2_ solution. A fixed-bed tubular reactor was filled with the IMs and then utilized for the removal of organic matter and nutrients from wastewater. Continuous aeration was sustained to maintain the aerobic conditions. The influent for the reactor comprised domestic wastewater sourced from the same treatment plant where the activated sludge was initially obtained.

A significant reduction in organic matter was observed, with the COD decreasing from 198.4 to 42.5 mg.L^−1^. Additionally, the total phosphorus concentration exhibited a decline from 2.48 to 0.14 mg.L^−1^ over a residence time of 12 h. Figure 7a illustrates the removal of COD and TP associated with the IMs. An exponential COD removal was observed, reaching 26.45% at 2 h, 55% at 6 h, and a maximum removal of 71.1% at a residence time of 12 h. TP also presented a gradual removal, reaching nearly 60% at 4 h and achieving 93.3% at 12 h.

Figure 7b depicts the kinetics of COD and TP degradation using Na-Alg beads alone and with IMs. The experimental results demonstrate that the reactions followed a first-order kinetic model for COD (R^2^ = 0.9705) and TP (R^2^ = 0.9983). Figure 7c shows the oxygen uptake rate trials, indicating that Na-Alg beads with IMs consumed oxygen and water pollutants. The oxygen saturation in the sample was 6.2 mg.L^−1^. Garcia-Ochoa et al. [154] noted that the DO concentration should be maintained at higher than 10% of the saturation value in order to ensure the microorganisms’ survival; therefore, the DO concentration was maintained at this level in the experiment.

Figure 7d displays the oxygen uptake rate (OUR) as a function of time. It can be observed that, after 12 h, complete DO depletion occurred in the sample with the immobilized consortium, which was likely due to its consumption by microorganisms. The relationship between DO and biomass concentration is linked through the oxygen mass balance [156], and the levels of oxygen consumption and OUR values are contingent upon the biomass quantity. The activated sludge was analyzed for volatile suspended solids (VSSs), and the biomass displayed values of approximately 5600 mg.L^−1^. This magnitude was notably lower when compared to the data reported in alternative studies [20,97]. The correlation between DO and biomass concentration is intricately connected through the oxygen mass balance, as elucidated by De la Morena et al. [152].

Other research studies have also highlighted the effectiveness of IMs in wastewater treatment. For instance, in the study conducted by Shin et al. [119], the physical and chemical characteristics of IMs (activated sludge) on polyvinyl alcohol were investigated. They reported an impressive 99% nitrification and de-nitrification, along with an 85% removal of organic matter. In another study by Banerjee et al. [68], the immobilization of *Chlorella vulgaris* on Na-Alg at 3% *w/v* demonstrated an efficient removal of nutrients (NH_4_^+^ and PO_4_^3−^) from wastewater. This suggests that immobilizing microorganisms on Na-Alg provides an effective method for the treatment of domestic wastewater, achieving significant removal rates for both organic contaminants and nutrients while concurrently minimizing sludge generation.

The use of sodium alginate in wastewater treatment brings significant advantages related to its biodegradable nature, water retention capability, and low toxicity. Despite these positive attributes, its implementation may encounter potential limitations, including higher costs due to manual processes, limited solubility under specific conditions, sensitivity to water characteristics, and competition with other organic substances. These limitations could impact the effectiveness of Na-Alg in specific environments. Therefore, significant attention has been paid to the development of hybrid materials—also known as coordination polymers [158,159]. Likewise, Ye et al. [160] carried out studies on the use of conjugated polymer photocatalysts for hydrogen production. Overall, water treatment using Na-Alg is an efficient and environmentally friendly method. This treatment could be complemented with the utilization of renewable energies, which can have positive impacts on environmental sustainability, including water management and treatment [161].

## 13. Challenges and Future Investigations

The use of Na-Alg in the immobilization of microorganisms for wastewater treatment presents challenges in key areas of research. The stability and durability of alginate matrices in adverse environments are noteworthy aspects, suggesting the need to enhance resistance through the development and use of complementary methods and materials. Moreover, crucial challenges concerning the optimization of treatment efficiency and the adaptability of IMs may be encountered. Future research should investigate the recyclability of alginate matrices to gain a better understanding of the microorganism–matrix interaction. Although combinations of natural and synthetic polymers have been explored without significant advantages being found, future research could delve into the associated interaction mechanisms, taking into consideration diverse environmental conditions. Additionally, promising areas of investigation include the development of new biomaterials, the exploration of innovative techniques using natural materials, and the use of hybrid materials such as coordination polymers. There is a notable interest in employing porous conjugated polymers as photocatalysts, potentially enhancing wastewater treatment through the immobilization of hybrid photocatalysts in collaboration with Na-Alg. Furthermore, the adaptation of microorganisms with nanoparticles in Na-Alg matrices has emerged as a promising option to improve performance. Additionally, an evaluation of the life cycles of polymers such as Na-Alg has been deemed essential for advancing the sustainability and effectiveness of immobilization technology. Research in this field will continue to drive commercial and field adaptations, demonstrating its crucial relevance in addressing pollution issues and achieving sustainable development goals.

## 14. Conclusions

This study comprehensively addressed the application of microorganism immobilization technologies in wastewater treatment, with a notable focus on Na-Alg as a key material in this process. Throughout this study, fundamental topics were explored, including the selection of the material used for immobilization, the intrinsic properties of polymers used in water treatment processes, the versatility and specific applications of Na-Alg, and the methods for microorganism immobilization. Furthermore, crucial aspects were examined in detail, such as nutrient degradation, the degradation kinetics of contaminants, and the dissolved oxygen consumption rate of IMs. Additionally, a case study was presented, demonstrating the effectiveness of microorganisms immobilized in Na-Alg for wastewater treatment, as high rates of contaminant removal and oxygen consumption were achieved by the microorganisms used in the process. Although promising results have been obtained, challenges persist, particularly regarding the stability of Na-Alg in adverse environments, underscoring the importance of future research focused on the sustainability and continuous improvement of this immobilization technology. These future investigations should be focused on recyclability, microorganism–matrix interactions, the development of new biomaterials, and the life cycle assessment of polymers such as Na-Alg in order to further promote the sustainability and effectiveness of immobilization technology.

## Figures and Tables

**Figure 1 polymers-16-00788-f001:**
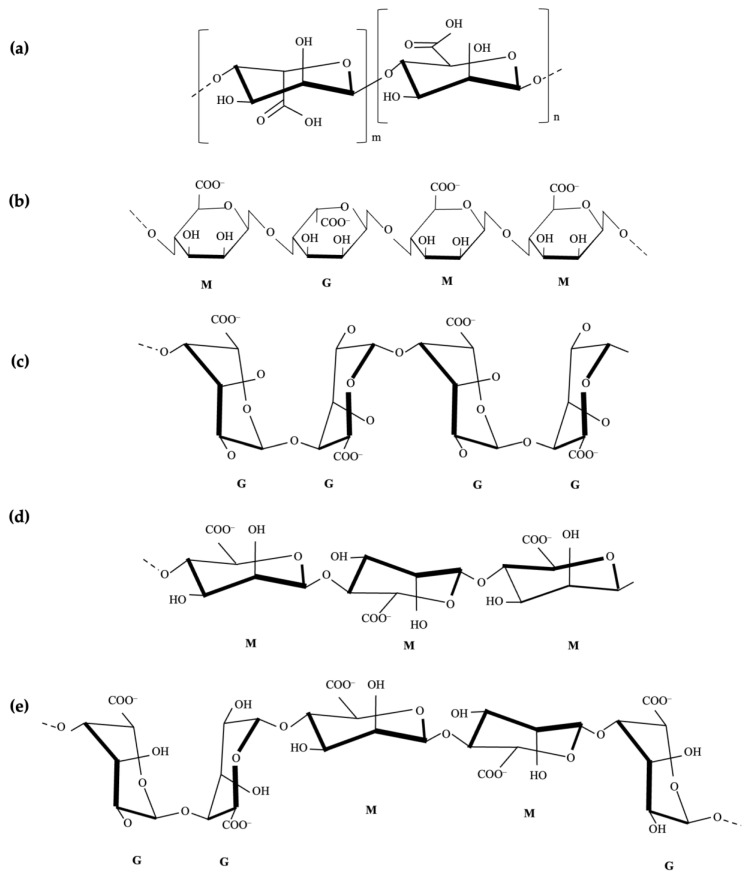
Sodium alginate structure (**a**). Chain segment structure of alginate (MGMM) (**b**). Blocks of acid groups in grouping (guluronicz) GG (**c**), (mannuronic) MM (**d**), and GGMMG (**e**).

**Figure 2 polymers-16-00788-f002:**
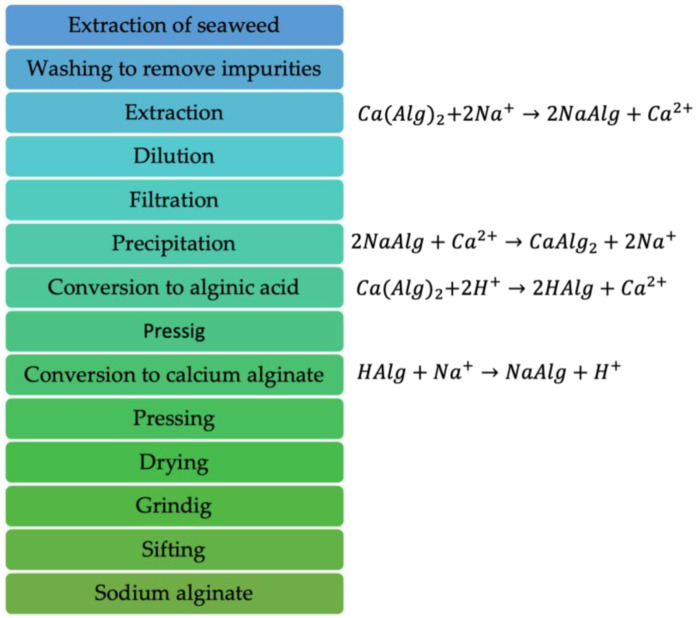
Synthesis to produce sodium alginate.

**Figure 3 polymers-16-00788-f003:**
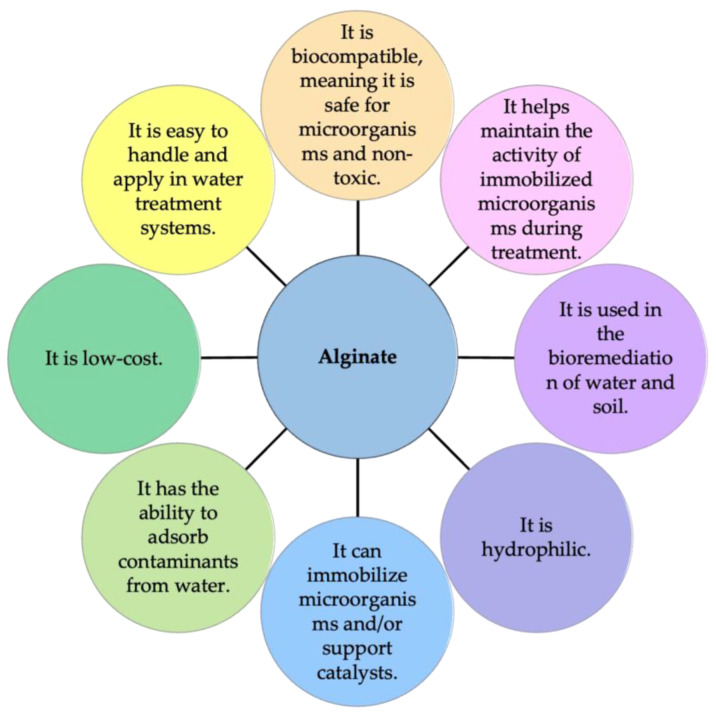
Advantages of sodium alginate in environmental re-measurement.

**Figure 4 polymers-16-00788-f004:**
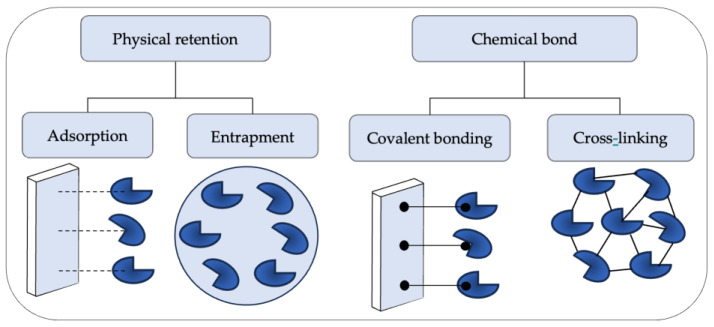
Methods for immobilizing microorganisms.

**Figure 5 polymers-16-00788-f005:**
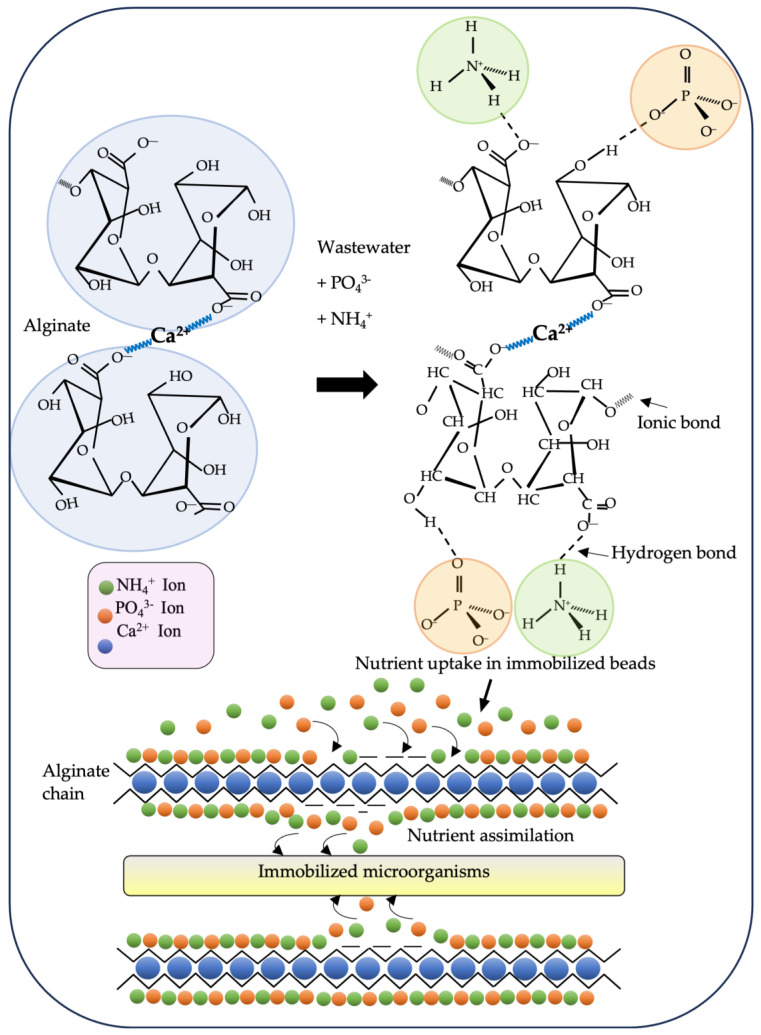
Mechanism of nutrient (NH_4_^+^ and PO_4_^3−^) adsorption and assimilation by alginate-trapped microorganisms.

**Figure 6 polymers-16-00788-f006:**
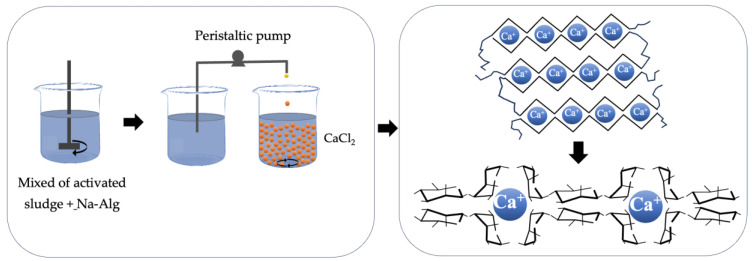
Schematic view of the preparation of Na-Alg beads with IMs.

**Figure 7 polymers-16-00788-f007:**
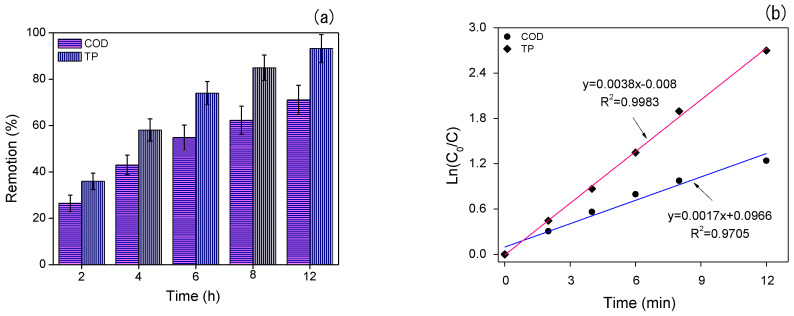
COD and TP degradation over time. (**a**) Removal percentage. (**b**) Removal kinetics. (**c**) Oxygen uptake rate (OUR) as a function of time. (**d**) Oxygen consumption by microorganisms.

**Table 1 polymers-16-00788-t001:** Studies that have used Na-Alg as an adsorbent.

Immobilization Support	Adsorbate	Adsorption	pH	Temperature	References
Hyperbranched polyamide/alginate	Sb(III)	195.7 mg/g	7	room temperature	[73]
Sodium alginate/biosilicate/magnetite	Malathion	36.76 mg/g	7	45 °C	[71]
Nanochitosan/sodium alginate	Pb(II)	178.57 mg/g	6.65	45 °C	[74]
Sodium alginate/activated charcoal	Cd (II)	137 mg/g	7	room temperature	[75]
Ethylenediamine/sodium alginate	Pb^2+^	219.3 mg/g	3–7	room temperature	[76]
Ethylenediamine/sodium alginate	Cu^2+^	87.8 mg/g	3–7	room temperature	[76]
Dolomite/sodium alginate	Phosphorous	28.45 mg/g	7–10	20 °C	[77]
Sodium alginate/ĸ-carrageenan	Antibiotic	291.6 mg/g	5.1	room temperature	[78]
Calcium alginate	Arsenic (V)	3.4 mg/g	4.5	room temperature	[79]
Calcium alginate	Phosphorus	19.42 mg/g	7.43	20 °C	[80]
Poly(vinyl alcohol)/sodium alginate	Phosphorus	107.53 mg/g	4	25 °C	[81]
Sodium alginate/polyethyleneimine	Methylene Blue	400 mg/g	5.5	60 °C	[82]
Acrylonitrile/sodium alginate	Methylene Blue	20.31 mg/g	7	70 °C	[83]
Poly(vinyl alcohol)-sodium Alginate/chitosan/montmorillonite	Methylene Blue	137.15 mg/g	8	room temperature	[84]

**Table 2 polymers-16-00788-t002:** Conditions that have been utilized for immobilizing microorganisms with Na-Alg.

Contaminant Removed	Na-Alg (*W*/*V*)	CaCl_2_	pH	Temperature	Removal	Reference
Phenol	2%	-	7	30 °C	100%	[85]
P-nitrophenol	1.4%	3%	5.5	55 °C	92%	[10]
Methyl blue	3%	5% *w*/*v*	5.5	30 °C	99%	[82]
Fluoride	1%	1% *w*/*v*	4.0	30 °C	92%	[86]
Phosphate	2%	3% *w*/*v*	3.0	30 °C	98.4%	[87]
Nutrient	3%	5.5%	7.0	27 °C	98%	[68]
Nutrient	2%	2% *w*/*v*	8.9	25 °C	95%	[88]
Pb^2+^	2%	2% *w*/*v*	5.0	25 °C	80%	[89]
Pb^2+^, Cu^2+^	1%	0.2 mol/L	4.5	25 °C	75.3%	[76]
Methyl blue	1%	0.1 mol/L	8.0	30 °C	90%	[84]
Phosphate	2%	2% *w*/*v*	4.5	45 °C	76%	[90]
Arsenic	1%	0.1 mol/L	4.5	25 °C	53.6%	[79]
Phosphorous	2%	2% *w*/*v*	8.5	23 °C	96%	[91]
Fluoride	2%	2% *w*/*v*	6.0	20 °C	90%	[92]
Copper	4%	4% *w*/*v*	3.0	21 °C	-	[93]
Uranium	1%	2% *w*/*v*	3.0	-	91%	[94]
Methyl blue	1%	22%	5.0	-	61%	[95]
Dyes	2%	2.2%	3.0	30 °C	95.4%	[96]
Ammonium	2%	5%	7.0	25 °C	70%	[97]

**Table 3 polymers-16-00788-t003:** Removed contaminants and removal efficiency using Na-Alg IMs.

Degraded Pollutant	%Removal	Microorganism	Immobilizing Matrix	References
Oil	99.73	E. Coli and aureus	Sodium alginate/chitosan–Ag	[128]
Nutrients	89	*Chlorella vulgaris*	Sodium alginate	[68]
Phenol	99.5	*Bacillus cereus*	Alginate/biochar	[12]
Diesel	68.68	*Halomonas* and *Aneurinibacillus*	Sodium alginate	[117]
Cadmium, arsenic, mercury, chromium	100	*E. coli*	Sodium alginate	[3]
COD and TP	94.26	Activated sludge	Sodium alginate	[19]
Ammonia	90	Activated sludge	Polyvinyl alcohol	[113]
Nutrients	95	*Chlorella vulgaris* and *Chlamydomonas reinhardtii*	Sodium alginate	[129]
P-nitrophenol	92	Activated sludge	Polyvinyl alcohol/alginate	[10]
Oil	70	*Pseudomonas*, *Bacillus*, *Acinetobacter*	Polyacrylamide/sodium alginate	[130]
Sulfate	99.4	Sulfate-reducing bacteria	Sodium alginate	[120]
Nitrogen	88	Denitrifying bacteria and anaerobic bacteria	Sodium alginate	[105]
Ammonia	96.5	*Bacillus subtilis*	Chitosan/sodium alginate	[56]
Nitrate and ammonia nitrogen	90.3	Ammonia-oxidizing bacteria	Polyvinyl alcohol/alginate	[131]
2,3′,4,4′,5-pentachlorodiphenyl	50.5	*Pseudomonas and Stenotrophomonas*	Sodium alginate	[25]
Nitrate	95	Activated sludge	Sodium alginate/kaolin	[13]
Ammoniacal nitrogen	89	Nitrifying bacteria	Sodium alginate	[16]
Nutrients and total organic carbon	96.5	*Chlorella pyrenoidosa*	Sodium alginate/biochar	[132]

**Table 4 polymers-16-00788-t004:** Adsorption isotherms used to assess IM treatments.

Isotherm		Terms	References
Langmuir	$q_{e}=\frac{q_{max}K_{L}C_{e}}{1+K_{L}C_{e}}$	*q_e_* is the adsorption amount per adsorbent unit mass in equilibrium (mg.g^−1^); *q_max_* is the maximum adsorption capacity per unit mass of adsorbent (mg g^−1^); *C_e_* is the pollutant equilibrium concentration in solution (mg.L^−1^); and *K_L_* is the Langmuir’s constant based on the affinity of the adsorbate binding site per adsorbent (L.g^−1^). *K_F_* and n are the Freundlich constants.	[68] [141]
Freundlich	qe=KFCe1n		[71] [89]
Redlich–Peterson	$q_{e}=\frac{K_{RP}C_{e}}{1+BC_{e}^{\beta }}$	*K_RP_* is the Redlich–Peterson isotherm constant (L.g^−1^), *B* is also a constant (L.mg^−1^), *ß* is an exponent that lies between 0 and 1, *C_e_* is the equilibrium liquid phase concentration of the sorbate (mg.L^−1^), and *q_e_* is the equilibrium sorbate loading by the sorbent (mg.g^−1^).	[140] [142]
Sips	qe=q0KsCe1n1+KsCe1n	*q_e_* (mmol·g^−1^) is the amount adsorbed at equilibrium, *C_e_* (mmol·L^−1^) is the equilibrium concentration of the adsorbate, *q_0_* (mmol·g^−1^) is the maximum adsorption capacity of Sips, *K_S_* is the Sips equilibrium constant, and *n* is the exponent of the Sips model related to the heterogeneity of the system.	[143] [144]
Temkin	$q_{e}=B_{1}ln\left(K_{T}\right)+B_{1}ln\left(C_{e}\right)$	*B_1_ = RT/b1* denotes the Temkin constant (J.mol^−1^), *R* is the universal gas constant (equal to 8.314 J/mol.K), *T* is the absolute temperature (°K), and *K_T_* and *b1* represent the equilibrium binding constant (L.g^−1^) and adsorption heat (kJ.mol^−1^), respectively.	[71]
Dubinin–Radushkevich	${ln\;q}_{e}=ln\left(q_{m}\right)-K\varepsilon^{2}$	*q_e_* is the equilibrium adsorbent phase concentration of adsorbate (mg.g^−1^), *q_m_* is the theoretical saturation capacity (mg.g^−1^), *K* is the activity coefficient related to the mean free energy of adsorption (mol^2^.kJ^−2^), and *ɛ* is the Polanyi potential (kJ.mol^−1^).	[145] [146]
Toth	qe=qsCebT+Cet1t	*q_s_* is the maximal adsorption, *t* is the Tóth parameter (reflecting the inhomogeneity of adsorbent), *q_eq_* is the equilibrium concentration on the solid phase (reflecting the adsorbate sequestering capacity), *C_e_* is the equilibrium concentration in the bulk fluid phase, *q_max_* is the maximal sequestering capacity in the mono-layer case, and *b_T_* is the Tóth constant (characterizing the interaction between the adsorbed and solution molecules).	[147]

## Data Availability

Data are contained within the article.

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
