# Peer review of "A Review of the Strategic Use of Sodium Alginate Polymer in the Immobilization of Microorganisms for Water Recycling"

_polymers, 2024, doi:10.3390/polym16060788_

Round 1

Reviewer 1 Report

Comments and Suggestions for Authors

The review paper offers some interesting and significant newer insights into the application of this assorted class of sodium alginate polymer for varied water recycling applications, and hence I reservations recommend its acceptance after addressing the main concerns:

1. When it deals with water recycling applications, authors should better explain how sodium alginate polymers have their unique advantages in these three aspects and provide some examples.

2. Please reframe the future perspective and conclusion section

3. Also, try to include one small conclusive remark after each main section. This will strengthen the review.

4. Some work on the regarding water pollution by using different materials, such as Sol. RRL, 2023,7, 2300143; Molecules 2023, 28, 4507 and Molecules 2023, 28, 6848.

5. Authors may look into it. Also, what are the pros and cons of sodium alginate polymer?

6. I feel texts and symbols in Figs have poor quality, please enhance if possible.

Comments on the Quality of English Language

check

Author Response

The review paper offers some interesting and significant newer insights into the application of this assorted class of sodium alginate polymer for varied water recycling applications, and hence I reservations recommend its acceptance after addressing the main concerns:

Dear reviewer, I want to express my sincere gratitude for the valuable feedback provided regarding this manuscript. I have carefully reviewed your observations and have implemented the following modifications in response to your suggestions. I am committed to enhancing the quality and understanding of this work. I appreciate the opportunity to improve and refine the work according to your guidance.

  1. When it deals with water recycling applications, authors should better explain how sodium alginate polymers have their unique advantages in these three aspects and provide some examples.

I deeply appreciate your detailed review and constructive comments on this work. Changes have been made to the document, providing an explanation of the unique advantages of sodium alginate polymers. Additionally, Figure 3 has been included, illustrating the advantages of alginate as a polymer for bioremediation.

  1. Please reframe the future perspective and conclusion section

I appreciate the reviewer's suggestion. The sections on future perspectives and conclusions have been rewritten. I hope that the proposed modifications meet expectations and contribute to strengthening the content of the document.

  1. Also, try to include one small conclusive remark after each main section. This will strengthen the review.

Thank you for this important recommendation. A brief concluding remark has been included after each main section.

  1. Some work on the regarding water pollution by using different materials, such as Sol. RRL, 2023,7, 2300143; Molecules 2023, 28, 4507 and Molecules 2023, 28, 6848.

I sincerely appreciate your suggestion to consider works related to water pollution using various materials. I have reviewed the recommended articles and have added them to the document. The following recommended references have also been added:

Huang, X. M., Chen, N., Ye, D. N., Zhong, A. G., Liu, H., Li, Z., & Liu, S. Y. (2023). Structurally Complementary Star‐shaped Unfused Ring Electron Acceptors with Simultaneously Enhanced Device Parameters for Ternary Organic Solar Cells. Solar RRL.

Ye, D.; Liu, L.; Peng, Q.; Qiu, J.; Gong, H.; Zhong, A.; Liu, S. Effect of Controlling Thiophene Rings on D-A Polymer Photocatalysts Accessed via Direct Arylation for

Zhao, J.; Dang, Z.; Muddassir, M.; Raza, S.; Zhong, A.; Wang, X.; Jin, J. A New Cd(II)-Based Coordination Polymer for Efficient Photocatalytic Removal of Organic Dyes. Molecules 2023, 28, 6848. https://doi.org/10.3390/molecules28196848

  1. Authors may look into it. Also, what are the pros and cons of sodium alginate polymer?

Thank you for this important recommendation. The recommended information has been added, including Figure 3.

  1. I feel texts and symbols in Figs have poor quality, please enhance if possible.

I sincerely appreciate your attention to this detail. I have carefully reviewed the mentioned figures, and I confirm that all figures meet the necessary quality standards. Thank you for bringing it to my attention.

I want to express my sincere gratitude for the valuable feedback provided regarding this manuscript. I have carefully reviewed your observations and have implemented the modifications in response to your suggestions. I appreciate the opportunity to improve and adjust the work according to your guidance.

Reviewer 2 Report

Comments and Suggestions for Authors The summary of few different papers is not good enough unless you summaries a finding from their collective findings and based on that should be able to make few suggestions/comments/recommendations of your own. 1. A summary of the main recent progresses and remaining challenges of the usual characterization techniques should be given including recent publications. 2. A sub-section should be created to discuss the intrinsic and specific properties/features of polymers that makes them suitable to be used as a new materials in improving water quality, preserving. 3. I suggest the author could give a sub-section should be created to provide general/specific synthesis procedure for polymer and then discussion on how these strategies influence the properties of the polymer.

4. Some refs may be considered, such as CrystEngComm, 2019, 21:4578-4585; Coord. Chem. Rev., 2020, 406, 213145; CrystEngComm, 2021, 23, 741–747

Comments on the Quality of English Language

revise

Author Response

The summary of few different papers is not good enough unless you summaries a finding from their collective findings and based on that should be able to make few suggestions/comments/recommendations of your own.

Thank you very much for your time spent reviewing this document, as well as your valuable recommendations regarding this manuscript. I have carefully reviewed your observations and have made the necessary modifications.

  1. A summary of the main recent progresses and remaining challenges of the usual characterization techniques should be given including recent publications.

I sincerely appreciate your attention to this detail. Text has been added to the section Sodium alginate as adsorbent, as well as to the last section.

  1. A sub-section should be created to discuss the intrinsic and specific properties/features of polymers that makes them suitable to be used as a new materials in improving water quality, preserving.

I sincerely appreciate your attention to this detail. I have added the requested information in subsection 3 (and two new references).

  1. I suggest the author could give a sub-section should be created to provide general/specific synthesis procedure for polymer and then discussion on how these strategies influence the properties of the polymer.

I appreciate the suggestion. I have taken into account the recommendations and have incorporated text and an additional image (Image 2) providing a synthesis procedure of the polymer in question.

  1. Some refs may be considered, such as CrystEngComm, 2019, 21:4578-4585; Coord. Chem. Rev., 2020, 406, 213145; CrystEngComm, 2021, 23, 741–747.

I sincerely appreciate your suggestion to consider other works related to various materials. I have reviewed the recommended articles and have added them to the document:

Pan, Y., Ding, Q., Xu, H., Shi, C., Singh, A., Kumar, A., & Liu, J. (2019). A new Zn (ii)-based 3D metal–organic framework with uncommon sev topology and its photocatalytic properties for the degradation of organic dyes. CrystEngComm, 21(31), 4578-4585.

Liu, J. Q., Luo, Z. D., Pan, Y., Singh, A. K., Trivedi, M., & Kumar, A. (2020). Recent developments in luminescent coordination polymers: Designing strategies, sensing application and theoretical evidences. Coordination chemistry reviews, 406, 213145.

Wang, J., Rao, C., Lu, L., Zhang, S., Muddassir, M., & Liu, J. (2021). Efficient photocatalytic degradation of methyl violet using two new 3D MOFs directed by different carboxylate spacers. CrystEngComm, 23(3), 741-747.

Reviewer 3 Report

Comments and Suggestions for Authors

1. I would recommend to mention source of microorganism, the plant details (Case Study section, raws 441-442).

2. Raw 491: I would recommend to include references as well after words "when compared with data reported in alternative studies".

3. In case authors repeated the experiments for statistics, would be good to include error bars in Figure 4a.

Author Response

Dear reviewer, I greatly appreciate your valuable comments on this manuscript. Your observations have been carefully reviewed, and the necessary modifications have been made.

  1. I would recommend to mention source of microorganism, the plant details (Case Study section, raws 441-442).

I appreciate your recommendation. In the new version of the document, the recommended information has been added.

  1. Raw 491: I would recommend to include references as well after words "when compared with data reported in alternative studies".

Thank you for your observation. The wording has been improved, and two references have been added.

  1. In case authors repeated the experiments for statistics, would be good to include error bars in Figure 4a.

I appreciate your recommendation. I do have statistics from the experiments; therefore, error bars have been added to Figure 4(a) accordingly.

Round 2

Reviewer 1 Report

Comments and Suggestions for Authors

accept

Reviewer 2 Report

Comments and Suggestions for Authors

it could be accepted based on its revision